# Towards Distortion-Debiased Blind Image Quality Assessment

## ABSTRACT

Existing blind image quality assessment (BIQA) models are susceptible to biases related to distortion intensity and domain. Intensity bias manifests as an over-sensitivity to severe distortions and underestimation of minor ones, while domain bias stems from the discrepancies between synthetic and authentic distortion properties. This work introduces a unified learning framework to address these distortion biases. We integrate distortion perception and restoration modules to address intensity bias. The restoration module uses a combined image-level and feature-level denoising method to restore distorted images, where easily restorable minor distortions serve as references for mildly distorted images, and severe distortions benefit directly from distortion perception. Finally, calculating a distortion intensity matrix via intensity-aware cross-attention for adaptive handling of intensity bias. To tackle domain bias, we introduce a distortion domain recognition task, leveraging inherent differences between synthetic and authentic distortions for adaptive quality score weighting. Experimental results show that our proposed method achieves state-of-the-art performance on a multitude of synthetic and authentic IQA benchmark datasets. The code and models will be available.

## CCS CONCEPTS

• **Computing methodologies → Computer vision tasks**; **Image processing**.

## KEYWORDS

Blind image quality assessment, Distortion bias, Pseudo reference, Distortion domain recognition

## 1 INTRODUCTION

The digital age has witnessed an exponential proliferation of visual information across diverse domains. However, this burgeoning volume of image data is susceptible to degradation during transmission and compression processes, necessitating the deployment of robust image quality assessment (IQA) algorithms. Blind IQA (BIQA) plays a crucial role in real-world scenarios by developing objective metrics that closely mimic human perception of distorted images in the absence of pristine reference images. With the advent of deep neural networks, deep learning-based BIQA approaches have achieved remarkable success by leveraging the powerful representational capabilities of these models to automatically extract informative features from distorted images. These approaches circumvent the

need for handcrafted feature engineering and facilitate accurate quality score prediction through learned mappings between the extracted features and subjective quality ratings.

In establishing deep learning-based BIQA models, two primary distortion domains are typically considered: synthetic distortions and authentic distortions. Synthetic distortions are artificially introduced under controlled conditions and encompass a range of common flaws with varying degrees and types, such as noise, blur, compression artifacts, and color shifts. Previous methods have primarily aimed to enhance model effectiveness in perceiving these types of distortions by identifying different distortion levels and categories. Distortion classification-based approaches [14, 25] build the model's capability to discern distinct distortion patterns by creating samples with various distortion types. Ranking-based methods enable models to learn the relative quality ordering of images with different distortion intensities [12, 15], while reference-based methods [2, 11, 16] attempt to reconstruct the pristine reference information of distorted images for subsequent quality regression. On the other hand, authentic distortions encompass a diverse array of intricate and often unpredictable degradation patterns that images may encounter in real-world scenarios, stemming from factors such as camera sensor noise, environmental interferences, or post-processing effects. To effectively handle the diversity of authentic distortions, semantic-aware BIQA models have demonstrated promising results, such as exploiting the rich statistical properties of semantic features [9], incorporating multi-scale semantic features [4, 21], and semantic representation of multi-aspect ratio images [6] to model the complex nature of authentic distortions.

Existing BIQA models are susceptible to distortion bias including distortion domain and intensity, rendering it difficult to effectively handle the diverse degradation in practical situations. Distortion-domain bias refers to the domain dichotomy between synthetic and authentic distortions, which originate from diverse image degradation scenarios exhibiting distinct distortion characteristics [25, 26]. This poses a considerable challenge for BIQA models to jointly characterize both forms of degradation. However, current research mostly focuses on constructing BIQA models tailored to specific distortion scenarios. For instance, BIQA models trained in extensive collections of synthetic distortion samples often struggle to achieve comparable performance on authentic distortion images [12, 23]. Conversely, semantic-aware BIQA models designed for authentic distortions may exhibit limited robustness to synthetic distortions. Hence, there remains a lack of integrated studies that consider both synthetic and authentic distortions in image quality assessment.

Furthermore, distortion-intensity bias refers to the phenomenon observed in BIQA models, wherein they exhibit reduced sensitivity toward subtle distortions while demonstrating proficiency in perceiving severe distortions. This bias arises from the inherent nature of deep learning architectures and their optimization objectives. The model's structural design emphasizes the extraction of high-level, global features from images, making it challenging to

capture features associated with minor distortions accurately. In addition, semantic-aware-based BIQA models, designed primarily for semantic recognition tasks, are required inherently less sensitive to distortions. These factors collectively contribute to inaccurate quality evaluations, especially for subtly distorted images.

This paper tackles the challenges of distortion domain and intensity bias to improve model performance. For distortion-domain bias, a domain classification task is employed for effective recognition between synthetic and authentic domain. By leveraging domain information, the model can automatically classify distortion domains and subsequently weight the quality evaluation results of the distorted image across domains based on their inter-domain similarity. For distortion intensity bias, prior studies have investigated corrective techniques, such as image-level restoration [11, 16] and feature-level restoration [2], for restoring mildly distorted images to a level approximating the original image. These methods demonstrate particular effectiveness in addressing subtle distortions. In contrast, severely distorted images are challenging to fully recover, yet exhibit prominent distortion that the model can readily identify. Consequently, to address distortion-intensity bias, a combined strategy of direct evaluation and restored reference information is employed for assessing both mildly and severely distorted images. To mitigate the issues of distortion domain and intensity bias, this work develops a novel blind image quality assessment framework. The main contributions of this work are as follows:

- We proposed a BIQA model for adaptive image quality evaluation across diverse distortion domains, effectively characterizing varying degradation severities for practical applications.
- To address distortion intensity bias, a combined approach using single image and restored reference image evaluation is employed for adaptive distortion intensity assessment. To provide effective reference information, we propose a denoising module that integrates image-level and feature-level methods.
- To mitigate distortion domain bias, a distortion domain discrimination task is incorporated to adaptively weight evaluation results based on inter-domain similarity.
- Experiments conducted on multiple IQA benchmark datasets including synthetic and authentic datasets demonstrate that the proposed model achieves state-of-the-art performance, yielding superior results on both domains of datasets.

## 2 RELATED WORKS

## 2.1 BIQA methods with recovered information

By comparing the differences between Full-Reference IQA (FRIQA) and BIQA, the reference-based BIQA method was proposed, which attempts to reconstruct the original reference information of the distorted image for quality assessment. Lin and Wang [11] used a stacked hourglass model to generate illusion images under the constraints of distorted images, and used an IQA discriminator to judge the quality of the illusion images. Finally, they fused the difference information between the illusion image and the distorted image with high-level semantics to guide the learning of the quality regression network. Chen et al. [2] proposed a pseudo-reference BIQA method based on the feature layer. They learned to obtain

pseudo-reference features from distorted images through a mutual learning scheme between distorted images and reference images, and finally obtained the final quality score through quality aggregation. Pan et al. [16] used a non-adversarial model to handle the task of restoring distorted images. They employed multiple visual compensation modules, composed of convolutional layers and pooling layers, and asymmetric residual blocks to manage the quality reconstruction relationship between distorted images and restored images. Finally, they fused multi-level restoration features with distorted image features to perform quality estimation. This approach allows for effective quality assessment and restoration of distorted images.

## 2.2 BIQA methods with single distorted image

When constructing BIQA models, two main domains of distortions are usually considered: synthetic distortion and authentic distortion. Some methods are designed for single distorted image. For synthetic distortions with clear distortion types and levels, BIQA models mainly use methods based on distortion classification or distortion ranking. Zhang et al. [25] use a method based on distortion classification, using convolutional neural networks to classify distortion levels and types, and then using bilinear merging to form a unified representation for quality evaluation. Liu et al. [12] use a method based on distortion ranking. They manually generate sequence images of different qualities, pre-train them through a twin network, and then extract image features. When dealing with complex authentic distortions, BIQA models mostly use semantic perception methods. Golestaneh et al. [4] use a mixture of convolutional neural networks and Transformer models based on self-attention, simultaneously extracting local features and non-local features of the image, and integrating multi-scale semantic features. Su et al. [21] use a local distortion-aware module to capture image quality, integrating local distortion features and global semantic features to aggregate fine-grained details and global information, and finally predict quality through multi-scale representation.

## 3 MOTIVATION

The performance of BIQA models is often hampered by the presence of distortion bias. This bias arises from various factors and manifests in two primary forms: distortion type bias and distortion intensity bias. These biases have distinct root causes and exert differing effects on IQA model performance. A detailed analysis of these biases is provided in the subsequent sections.

## 3.1 Distortion-intensity bias

Distortion-intensity bias is typically caused by the varying sensitivity of IQA models to distortion intensities. Current BIQA models primarily leverage pre-trained semantic models on large-scale datasets, which exhibit robust deep semantic representation capabilities that are insensitive to minor noise, thereby hindering the accurate detection of subtle distortion variations. As illustrated in Figure 1 (a), we visualize the absolute prediction errors of two BIQA models, namely DB-CNN (utilizing a CNN backbone) [25] and DEIQT (utilizing a ViT backbone) [19], across varying distortion levels on the KAdid10k dataset, with five random experimental runs and an average result. In Figure 1 (a), it is evident that when the

 

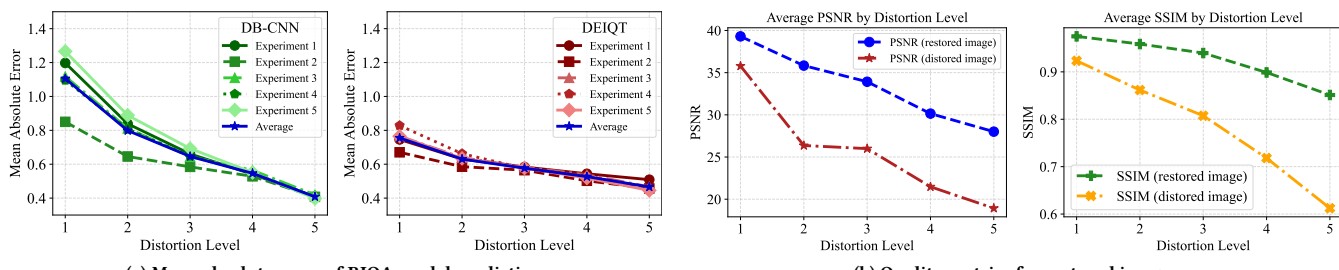

(a) Mean absolute error of BIQA model predictions

(b) Quality metrics for restored images

Figure 1: An analysis of prediction bias at different distortion intensities. (a) The average absolute error of BIQA model predictions across different distortion levels decreases as the distortion level increases, indicating the existence of an intensity bias where the models are insensitive to minor distortions. (b) The lower the distortion level, the better the quality of the restored images, suggesting that low distortion levels are more amenable to restoration compared to higher levels.

distortion intensity is low, the models with pre-trained semantic model exhibit a larger average absolute error between their scores and the ground truth. As the distortion level increases, the models' prediction errors tend to diminish. It is worth noting that although DEIQT outperforms DB-CNN, it still exhibits the phenomenon caused by the intensity bias.

However, mild distortions can be effectively mitigated through image restoration techniques, achieving quality levels close to the original images , and reference information can help the model learn the details of slight distortions. Figure 1 (b) showcases the PSNR and SSIM values of the distorted images and the images restored by the proposed image restoration model. The lower the distortion level, the closer the restored image quality approximates the original, resulting in superior PSNR and SSIM values. As the distortion intensity increases, the distortion features become more prominent, posing a challenge for the model to recover the original quality, leading to inaccurate reference information from the restoration process. With the results from Figure 1 (a), the evaluation of highly distorted images should benefit from individual distorted images, while mildly distorted images can be effectively restored and further utilized as reference information to enhance the accuracy of the evaluation.

## 3.2 Distortion-domain bias

Distortion-domain bias stems from inherent discrepancies between distortion domains encountered in BIQA tasks, commonly comprising synthesized and authentic distortions with fundamentally distinct statistical profiles. BIQA models trained exclusively on a single distortion dataset lack generalizability and exhibit precipitous performance drops when evaluated on other distortion datasets due to domain shifts. As evidenced in Figure 2, we visualize the cross-dataset testing results of two BIQA models (DB-CNN [25] and PQR [24]). The left plot shows models trained on the authentic dataset (LIVEC [3]) achieving markedly better results on the sizable authentic KonIQ-10k [5] dataset compared to three synthetic datasets, highlighting sensitivity to the training domain. Meanwhile, the right plot indicates models trained on TID2013 [18] experience severe accuracy degradation when tested on the authentic LIVEC relative to synthetic datasets (LIVE [20] and CSIQ [8]). This indicates that image quality assessment models trained on datasets

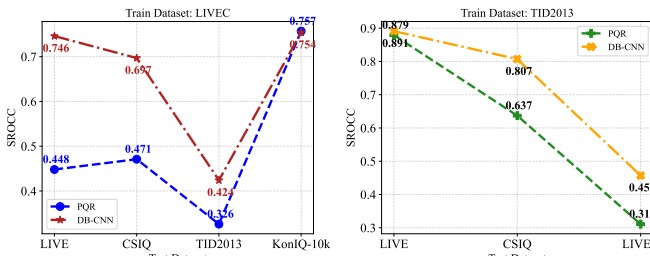

Figure 2: An analysis of distortion domains bias in BIQA model. Left: BIQA models trained on the authentic dataset (LIVEC) excel on the authentic dataset KonIQ-10k but struggle with synthetic datasets (LIVE, CSIQ, and TID2013). Right: BIQA models trained on the synthetic dataset (TID2013) perform well on synthetic datasets (LIVE and CSIQ) but degrades significantly on the authentic dataset (LIVEC).

of the same distortion domain struggle to generalize well when confronted with datasets from different domains. These findings underscore the necessity of accounting for diverse distortion domains in BIQA modeling to formulate domain-aware quality assessment strategies attenuating such biases.

## 4 THE PROPOSED METHOD

The proposed framework is illustrated in Figure 3. To address distortion-intensity bias, an image restoration module is first developed to infer the latent pristine reference form. Then, feature-level reference information is constructed by extracting the pixel-wise discrepancy features between the restored and distorted images to represent distortion intensity, with further supervision from the ground-truth to precisely represent the distortion variations. Intensity-aware cross-attention exploits discrepancy features to achieve interaction between distorted features and reference information. Regarding distortion domain bias, a distortion domain recognition task is introduced to adaptively weight evaluation outcomes across domains according to measured inter-domain similarity. Furthermore, multi-scale feature extraction and enhancement

of the feature spatial layout and channels are conducted to enhance representational capacity. The detailed method is outlined as follows.

## 4.1 Debiased modeling of distortion intensity

We first employ image restoration techniques to recover distorted images to a quality level approximating the original images, further utilizing them as references to enhance the model's robustness against mild distortions. We integrate image-level and feature-level restoration methods.

**Image Restoration Module (IRM):** In the initial stage, we construct an image eestoration module (IRM). The IRM utilizes a Swin-Transformer-based U-Net architecture. Details regarding Swin-UNet can be found in the cited reference [1]. The IRM is trained for image denoising using a combination of $\ell_1$ loss and structural similarity index (SSIM) loss. The pixel-wise $\ell_1$ loss is defined as the mean absolute difference between the processed image $\mathbf{x}$ and the ground truth $\mathbf{y}$ over the entire image domain,

$$\ell_1(\mathbf{p}) = \frac{1}{|\mathbf{p}|} \sum_{(i,j) \in \mathbf{p}} ||\mathbf{x}(i,j) - \mathbf{y}(i,j)||, \quad (1)$$

where $\mathbf{x}(i,j)$ and $\mathbf{y}(i,j)$ correspond to the pixel values of the processed image and the ground truth at location $(i,j)$ respectively, and $|\mathbf{p}|$ signifies the total number of pixels within the image patch $\mathbf{p}$. The SSIM is computed as follows,

$$\ell_{ssim}(\mathbf{p}) = \frac{(2\mu_{\mathbf{x}}\mu_{\mathbf{y}} + C_1)}{(\mu_{\mathbf{x}}^2 + \mu_{\mathbf{y}}^2 + C_1)} \cdot \frac{(2\sigma_{\mathbf{xy}} + C_2)}{(\sigma_{\mathbf{x}}^2 + \sigma_{\mathbf{y}}^2 + C_2)}, \quad (2)$$

where $\mu_{\mathbf{x(p)}}$ and $\mu_{\mathbf{y(p)}}$ denote the local means of the processed image $\mathbf{x}$ and ground truth image $\mathbf{y}$, respectively, computed over the image patch $\mathbf{p}$. $\ell_{ssim}(\mathbf{p})$ quantifies the structural similarity between $\mathbf{x}$ and $\mathbf{y}$ including luminance, contrast, and structural information. The IRM is optimized by jointly minimizing the $\ell_1$ and SSIM losses by,

$$\ell_{res} = \alpha \cdot (1 - \ell_{ssim}(\mathbf{p})) + (1 - \alpha) \cdot \ell_1(\mathbf{p}) \quad (3)$$

Subsequently, the distorted image $\mathbf{x}$, the restored image $\hat{y}$, and the original image $\mathbf{y}$ are jointly fed into a feature extraction module (FEM), represented by a parametric function $f$ with parameters $\theta_e$, to obtain their respective embedding representations as follows,

$$\mathcal{E}_{dis} = f(\mathbf{x}; \theta_e); \ \mathcal{E}_{res} = f(\hat{y}; \theta_e); \ \mathcal{E}_{ref} = f(\mathbf{y}; \theta_e) \quad (4)$$

**Discrepancy Perception Module (DPM):** Then, we quantify the intensity of image distortion by measuring the discrepancy between the restored image and the distorted image, where a larger discrepancy indicates a more severe level of distortion. We introduce a DPM, represented by a function $\psi$ with parameters $\theta_d$. The extracted features are further enhanced through multi-scale feature enhancement (MSFE), which will be elaborated in the next section, to optimize the spatial and channel information. The discrepancy representations are formulated as,

$$d_{res} = \psi(\mathcal{E}_{res} - \mathcal{E}_{dis}; \theta_d); \quad d_{ref} = \psi(\mathcal{E}_{ref} - \mathcal{E}_{dis}; \theta_d), \quad (5)$$

where $d_{res}$ and $d_{ref}$ represent the distortion levels of the restored and original images, respectively. The function $\psi$ learns to characterize the distortion discrepancy guided by the original image by

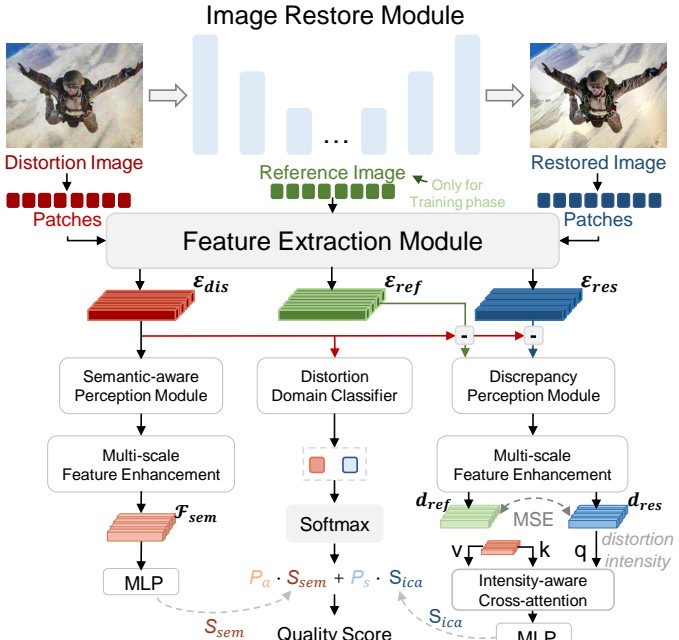

**Figure 3: An overview of the proposed distortion-debiased BIQA model.**

the following loss,

$$\ell_{\text{mse}}(i,j) = \frac{1}{|\Omega|} \sum_{(i,j) \in \Omega} ||d_{res}(i,j) - d_{ref}(i,j)||_2^2, \quad (6)$$

where $(i,j)$ are value coordinates in the feature domain $\Omega$. This loss encourages consistency between $d_{res}$ and $d_{ref}$, enabling the DPM to effectively capture distortion levels.

**Intensity-aware Cross-attention (ICA):** To adaptively weight the distortion features and restoration features according to the distortion intensity, we employ a cross-attention mechanism to generate feature weight matrices. Specifically, we use the intensity representation $d_{res}$ as the query $Q$, and the distortion features $\mathcal{F}_{sem}$ as the keys $K$ and values $V$, computing the attention weights via,

$$\mathbf{Q} = d_{res} \cdot W_{\mathbf{Q}}; \ \mathbf{K} = \mathcal{F}_{sem} \cdot W_{\mathbf{K}}; \ \mathbf{V} = \mathcal{F}_{sem} \cdot W_{\mathbf{V}}, \quad (7)$$

$$\text{Attention}(\mathbf{Q}, \mathbf{K}, \mathbf{V}) = \text{softmax}\left(\frac{\mathbf{Q}\mathbf{K}^T}{\sqrt{d_k}}\right)\mathbf{V}, \quad (8)$$

where $W_Q$, $W_K$, $W_V$ are linear projection weight matrices, and $d_k$ is a scaling factor. The attention weight matrix is obtained via softmax and multiplied with $\mathbf{V}$ to yield the weighted feature representation. Finally, the ICA output features are passed through an MLP with two linear layers to output the predicted quality score $S_{ica}$.

## 4.2 Debiased modeling of distortion domain

The bias in distortion domains is often attributed to the inherent distribution discrepancy between authentic distortions and synthetic distortions. To address this issue, we first aim to identify the distortion domain of the image and then devise appropriate evaluation strategies accordingly.

**Distortion Domain Classifier (DDC):** Then, we propose a DDC in our model, represented by a function $\varphi$ with parameters $\theta_c$. The DDC takes $\mathcal{E}_{dis}$ as input and outputs two values representing the class probabilities for the synthetic distortion domain and the authentic distortion domain, respectively, formulated as,

$$[p_s, p_a] = \text{softmax}\left(\varphi(\mathcal{E}_{dis}; \theta_c)\right), \tag{9}$$

where $p_s$ and $p_a$ denote the probabilities of the input belonging to the synthetic and authentic distortion domains, respectively. The DDC is optimized using the following cross-entropy loss,

$$\ell_{cls} = -\mathbb{E}_{\mathbf{x} \sim \mathcal{D}_s} \log p_s(\mathbf{x}) - \mathbb{E}_{\mathbf{x} \sim \mathcal{D}_a} \log p_a(\mathbf{x}), \tag{10}$$

where $\mathcal{D}_s$ and $\mathcal{D}_a$ represent the synthetic and authentic distortion domains, respectively. By minimizing $\ell_{cls}$, the DDC is trained to accurately classify the distortion domain of the input image.

**Semantic-aware Perception Module (SPM):** The quality assessment of authentic distorted images benefits from pre-trained semantic-aware models trained on authentic distortion datasets (e.g., ImageNet), enabling precise capture of the complexities in authentic distortions and content. We introduce a semantic-aware perception module to handle authentic distortion scenarios, outputting $\mathcal{F}_{sem}$ through MFSE, which is further processed by an MLP to produce a quality score $S_{sem}$.

However, for mild distortions, the semantic-oriented perceptual models are not sufficiently sensitive. In addition, synthetic distortions, generated through specific distortion algorithms with diverse domains and levels, typically exhibit advantages in quality prediction for distortion restoration and distortion intensity awareness. To leverage the strengths of both assessment approaches, we dynamically weight the evaluation results from the two distortion domains based on their distortion domain similarity, computed as the following,

$$S_{total} = p_a \cdot S_{sem} + p_s \cdot S_{ica}, \tag{11}$$

where $p_a$ and $p_s$ are adaptive weighting factors determined by the distortion domain similarity.

## 4.3 Feature extraction and enhancement

The FEM employs the first six transformer blocks of the VIT-S/16 architecture [7]. The remaining modules, SPM, DPM, and DDC, are primarily composed of transformer blocks as well. The SPM and DPM have the same network structure, each consisting of six transformer blocks, while DDC comprises three transformer blocks. All modules are initialized with pre-trained weights from ImageNet. denoted as $z_1$, $z_2$, and $z_3$. These features are first concatenated: $z = z_1 + z_2 + z_3$, where + represents the concatenation operation. To enhance the representational capacity, we apply feature channel self-attention (FCSA) and spatial convolution on $z$. The calculation process is as follows,

$$\bar{z} = \text{F}(\text{Conv}(\text{R}(\text{FCSA}(z^\top \cdot W_\mathbf{Q}, z^\top \cdot W_\mathbf{K}, z^\top \cdot W_\mathbf{V})^\top))), \tag{12}$$

where $W_\mathbf{Q}$, $W_\mathbf{K}$, and $W_\mathbf{V}$ are linear projection parameters. where $\text{F}(\cdot)$ denotes the operation of flattening the feature maps into a feature sequence, $\text{R}(\cdot)$ represents the operation of reshaping the feature sequence into feature maps, and $^\top$ denotes the transpose operation. The convolution projects the concatenated features to

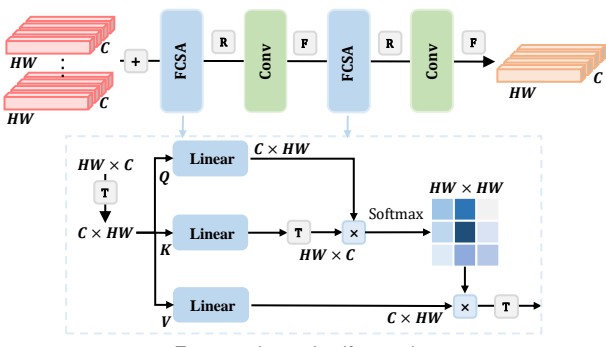

Feature channel self-attention

**Figure 4: The process of multi-scale feature enhancement. It involves utilizing feature channel self-attention (FCSA) and spatial convolution on the multi-scale features. In the figure, $\text{F}(\cdot)$ signifies the operation of flattening the feature maps into a feature sequence, $\text{R}(\cdot)$ indicates the operation of reshaping the feature sequence back into feature maps, the symbol + represents the concatenation operation, and $^\top$ represents the transpose operation.**

the model dimension. Followed by a second FCSA and convolution layer to obtain an enhanced feature representation,

$$z_{enh} = \text{F}(\text{Conv}(\text{R}(\text{FCSA}(\bar{z}^\top \cdot W_Q, \bar{z}^\top \cdot W_K, \bar{z}^\top \cdot W_V)^\top))) \tag{13}$$

The IRM module is separately trained while other modules are optimized end-to-end on IQA datasets using mean squared error (MSE) loss, with frozen IRM parameters.

## 5 EXPERIMENTAL RESULTS

### 5.1 Experimental Setups

**Datasets and Protocols:** Our experiments will be conducted on six IQA benchmark databases, LIVE [20], CSIQ [8], TID2013 [18], and KADID-10K [10] are synthetic distortion databases, and LIVEC [3] and KonIQ-10K [5] are authentic distortion databases. These databases provide subjective quality scores as a reference. In IQA, in order to compare the performance of models, two correlation criteria are often used, namely the Spearman rank correlation coefficient (SROCC) and the Pearson linear correlation coefficient (PLCC), and they are defined as,

$$\text{PLCC} = \frac{\sum_{i=1}^{n}(x_i - \bar{x})(y_i - \bar{y})}{\sqrt{\sum_{i=1}^{n}(x_i - \bar{x})^2 \sum_{i=1}^{n}(y_i - \bar{y})^2}}, \tag{14}$$

$$\text{SROCC} = 1 - \frac{6 \sum d_i^2}{n(n^2 - 1)}, \tag{15}$$

where $x_i$ and $y_i$ are the individual sample points indexed with $i$, $\bar{x}$ and $\bar{y}$ are the means of the sample points, $n$ is the total number of samples. $d_i$ is the difference between the ranks of the $i$-th pair of data, $n$ is the total number of samples.

**Implementation Details:** The framework we used is pytorch, and the GPU used for training is NVIDIA Tesla A800. For model training, we use the AdamW optimizer with the batch size and initial learning rate set to 8 and 1e-5, using the LambdaLR learning rate regulator.

**Table 1: Performance comparison with the state-of-the-art BIQA methods on six benchmark databases.**

| Methods | LIVE[20] | | CSIQ[8] | | TID2013[18] | | KADID-10k[10] | | LIVEC[3] | | KonIQ-10k[5] | |
|---|---|---|---|---|---|---|---|---|---|---|---|---|
| | SROCC | PLCC | SROCC | PLCC | SROCC | PLCC | SROCC | PLCC | SROCC | PLCC | SROCC | PLCC |
| *BIQA methods with recovered information:* | | | | | | | | | | | | |
| AIGQA [13] | 0.960 | 0.957 | 0.927 | 0.952 | 0.871 | 0.893 | 0.864 | 0.863 | 0.751 | 0.761 | - | - |
| FPR [2] | 0.967 | 0.968 | 0.948 | 0.956 | 0.872 | 0.887 | 0.901 | 0.899 | - | - | - | - |
| VCRNet [16] | 0.973 | 0.974 | 0.943 | 0.955 | 0.846 | 0.875 | - | - | 0.856 | 0.865 | 0.894 | 0.909 |
| DACNN [17] | 0.978 | 0.980 | 0.943 | 0.957 | 0.871 | 0.889 | 0.905 | 0.905 | 0.866 | 0.884 | 0.901 | 0.912 |
| CDINet [28] | 0.977 | 0.975 | 0.952 | 0.960 | 0.898 | 0.908 | 0.920 | 0.919 | 0.865 | 0.880 | 0.916 | 0.928 |
| *BIQA methods with single distorted image:* | | | | | | | | | | | | |
| DB-CNN [25] | 0.968 | 0.971 | 0.946 | 0.959 | 0.816 | 0.865 | 0.851 | 0.856 | 0.851 | 0.869 | 0.875 | 0.884 |
| CLRIQA [15] | 0.977 | 0.980 | 0.915 | 0.938 | 0.837 | 0.863 | 0.837 | 0.843 | 0.832 | 0.866 | 0.831 | 0.846 |
| HyperIQA [21] | 0.962 | 0.966 | 0.923 | 0.942 | 0.804 | 0.839 | 0.852 | 0.844 | 0.859 | 0.882 | 0.906 | 0.917 |
| TReS [4] | 0.969 | 0.968 | 0.922 | 0.942 | 0.863 | 0.883 | 0.859 | 0.858 | 0.846 | 0.877 | 0.915 | 0.928 |
| DEIQT [19] | 0.980 | 0.982 | 0.946 | 0.963 | 0.892 | 0.908 | 0.889 | 0.887 | **0.875** | 0.894 | 0.921 | 0.934 |
| Ours | **0.981** | **0.984** | **0.969** | **0.976** | **0.927** | **0.940** | **0.956** | **0.957** | 0.873 | **0.895** | **0.923** | **0.938** |

Each training image was cropped to a size of 224×224 and trained for a total of 300 epochs. For the BIQA methods compared in the paper, we directly use the performance in the original paper, and the untested datasets in the original paper we will derive from the source code replication provided by the authors.

**Training IRM:** To pre-train the IRM for synthetic distortions, we randomly selected 50,000 original images from KADIS-700K [10] and generated distorted images using the same 25 domains of distortions and 5 levels of distortion intensity as KADID-10K [10]. Each original image can yield 125 distorted samples. We used the AdamW optimizer, with the batch size and the initial learning rate set to 10 and 1e-4, respectively, and the cosine annealing learning rate regulator. Each training image was cropped to a size of 384×384 and trained for a total of 40 epochs. 50,000 images were randomly selected as the test set .

**Training DDC:** To pre-train the DDC, we selected 125000 images from the previously synthesized dataset as synthetic distortions and 125000 images from the COCO dataset as authentic distortions to compose our classification dataset. We use the pre-trained ViT-S model as our feature extraction module. We used the AdamW optimizer, with the batch size and the initial learning rate set to 10 and 1e-4, respectively, and the cosine annealing learning rate regulator. Each training image was cropped to a size of 384×384 and trained for a total of 40 epochs. 50,000 images were randomly selected as the test set .

## 5.2 Performance on Individual Databases

Based on the experimental setup of previous BIQA methods, we divided the dataset according to the ratio of 80% for training and 20% for testing. Among them, we divided the synthetic distortion dataset according to the reference image and divided the authentic distortion dataset directly. Ten random divisions are used for each experiment and the median result is selected.

To validate our model, we selected 10 state-of-the-art BIQA methods for comparison. They are mainly divided into two types of methods: BIQA methods with recovery information and BIQA methods with single distorted image. Table 1 shows the overall performance

**Table 2: SROCC evaluation on cross-datasets with different dataset domains.**

| Training | LIVEC [3] | |
|---|---|---|
| Testing | LIVE [20] | CSIQ [8] |
| TReS [4] | 0.531 | 0.562 |
| HyperIQA [21] | 0.570 | 0.581 |
| CLRIQA [15] | 0.624 | 0.652 |
| DB-CNN [25] | 0.746 | 0.679 |
| VIPNet [22] | 0.734 | 0.635 |
| Ours | **0.876** | **0.709** |

comparison on six benchmark databases, where the higher the SROCC and PLCC values, the better the performance.

Our model achieved state-of-the-art performance on five datasets, including LIVE, CSIQ, KADID-10K, TID2013, and KonIQ-10K. Although it did not achieve the most advanced performance on the LIVEC dataset, it is on par with the current state-of-the-art BIQA methods, showing highly competitive results. On KADID-10K, our SROCC and PLCC reached 0.956 and 0.957, respectively, which are 3.9% and 4.1% higher than the second-ranked CDINet. On TID2013, our SROCC and PLCC reached 0.927 and 0.940, respectively, an increase of 3.2% and 3.5% compared to CDINet, indicating that our method has a good effect in solving the distortion intensity bias commonly present in synthetic distortions. Compared with other BIQA methods with recovery information, it can be proven that our proposed recovery method combining image-level and feature-level has superiority. Compared with BIQA methods with single distorted image that cannot achieve good performance simultaneously on synthetic distortion and authentic distortion datasets, our proposed method of using a distortion domain classifier to adaptively weight quality scores achieves good results on both synthetic distortion datasets and authentic distortion datasets. In addition, the results in the table can also prove that regardless of the size of the data, the model can achieve good results, indicating that the model has

**Table 3: Performance results on mixed-dataset evaluations**

| Training | TID2013 [18] & LIVEC [3] | | | | | | | |
|---|---|---|---|---|---|---|---|---|
| Testing | LIVE[20] | | KADID-10K[10] | | KonIQ-10K[5] | | Average | |
| Methods | SROCC | PLCC | SROCC | PLCC | SROCC | PLCC | SROCC | PLCC |
| DB-CNN [25] | 0.900 | 0.883 | 0.544 | 0.544 | 0.620 | 0.606 | 0.688 | 0.678 |
| UNIQUE [27] | 0.899 | 0.872 | 0.514 | 0.466 | 0.663 | 0.625 | 0.692 | 0.654 |
| VCRNet [16] | 0.884 | 0.864 | 0.614 | 0.592 | 0.606 | 0.570 | 0.701 | 0.675 |
| SAWAR [29] | 0.903 | 0.890 | 0.657 | 0.647 | 0.665 | 0.634 | 0.742 | 0.724 |
| (Ours) | **0.949** | **0.942** | **0.781** | **0.774** | **0.765** | **0.822** | **0.832** | **0.846** |

**Table 4: Ablation studies on quality weighting strategy**

| Datasets | LIVE[20] | | CSIQ[8] | | TID2013 [18] | | LIVEC [3] | | Average | |
|---|---|---|---|---|---|---|---|---|---|---|
| Methods | SROCC | PLCC | SROCC | PLCC | SROCC | PLCC | SROCC | PLCC | SROCC | PLCC |
| $S_{sem}$ | 0.978 | 0.980 | 0.957 | 0.969 | 0.920 | 0.934 | 0.871 | 0.894 | 0.932 | 0.944 |
| $S_{ica}$ | 0.979 | 0.981 | 0.959 | 0.969 | **0.929** | **0.942** | 0.857 | 0.883 | 0.931 | 0.944 |
| Intensity-based weighting | 0.968 | 0.970 | 0.926 | 0.923 | 0.920 | 0.930 | 0.860 | 0.882 | 0.919 | 0.926 |
| Domain-based weighting | **0.981** | **0.984** | **0.969** | **0.976** | 0.927 | 0.940 | **0.873** | **0.895** | **0.938** | **0.949** |

high data efficiency. In summary, our model achieves the best performance or highly competitive performance on all six benchmark datasets.

**Table 5: Ablation studies on Multi-scale Feature Enhancement.**

| Datasets | MSFE | SROCC | PLCC |
|---|---|---|---|
| LIVE [20] | × | 0.979 | 0.981 |
| | ✓ | 0.981 | 0.984 |
| CSIQ [8] | × | 0.966 | 0.971 |
| | ✓ | 0.969 | 0.976 |
| TID2013 [18] | × | 0.918 | 0.932 |
| | ✓ | 0.927 | 0.940 |
| Average | × | 0.954 | 0.961 |
| | ✓ | 0.959 | 0.967 |

## 5.3 Performance on cross-datasets

To validate whether the impact of distortion domain bias has been reduced, we conducted two cross-dataset tests. The first cross-datasets experiment was to verify the generalization ability of a model trained on one domain of distortion dataset on another domain of distortion dataset. We trained the model on the authentic distortion dataset LIVEC, and then tested it on the synthetic distortion datasets LIVE and CSIQ. The results are shown in Table 2, with only SROCC reported. The results of TReS, HyperIQA, CLRIQA, and DBCNN come from VIPNet [22]. As can be seen from the table, our model achieved the best performance, leading the second place by 17.4% and 4.4% on the LIVE and CSIQ datasets respectively. This result indicates that our model has good generalization performance in cross-distortion domain experiments.

In the second cross-datasets experiment, to validate whether our adaptive scoring weighting method using a distortion domain

classifier can achieve excellent performance for both authentic and synthetic distortions, we mixed the authentic distortion dataset LIVEC with the synthetic distortion dataset TID2013 for training. We then tested the performance on CSIQ, KADID-10K, Koniq-10K, and LIVE. The results are shown in Table 3 3. The results of DB-CNN, UNIQUE [27], VCRNet, and SAWAR [29] come from SAWAR. As can be seen from the table, our model achieved the best performance, with improvements of 10%-30% on the KADID-10K and Koniq-10K datasets, and improvements of 5.1% and 5.8% in SROCC and PLCC on the LIVE , respectively. This result indicates that our model can handle the mixed situation of authentic and synthetic distortions well, and our proposed adaptive scoring weighting method is very effective. These two experiment results have proven that we have effectively solved the problem of distortion domain bias.

## 5.4 Ablation Studies

This section presents a series of ablation experiments designed to validate the effectiveness of individual model components. For all ablation experiments, the model is run ten times randomly, and the median results are reported. Below is the analysis of the ablation effects for each component:

**Distortion Domain Classifier (DDC):** To validate the effectiveness of DDC, we tested the results using only the distortion score and only the fusion score, respectively. Furthermore, to validate the effectiveness of the weights provided by the DDC module, we utilize $d_{res}$, representing the intensity of distortion, as input to obtain the weighted sum of the two scores, denoted as "Intensity-based weighting" in Table 4.

As shown in Table 4, the experimental results show that the fusion scores of most synthetic distortion datasets have achieved impressive results, and their performance is better than the distortion scores, indicating that we have effectively solved the problem of distortion intensity bias. For the authentic distortion dataset LIVEC, which is not greatly affected by distortion intensity bias,

the distortion score is better than the fusion score. The average results indicate that our weighted scores outperform the single scores, and our weighting effect is superior to diff weights. This validates the effectiveness of our method. This suggests that our approach of using a weighted combination of different scores and our specific weighting method can provide a more accurate and robust assessment.

**Multi-scale Feature Enhancement (MSFE):** As illustrated in Table 5, we conduct ablation experiments on MSFE. On the three datasets LIVE, CSIQ, and TID2013, incorporating MSFE results in an improvement of 0.52% and 0.62% in SROCC and PLCC in the average results, respectively, demonstrating the effectiveness of our proposed module.

## 6 CONCLUSION

Existing BIQA models are susceptible to intensity bias and distortion domain bias, where intensity bias refers to models being overly sensitive to severe distortions while underestimating mild distortions, and distortion domain bias indicates the inability of models to effectively generalize across different distortion domains. In this work, we design a distortion-debiased BIQA method to address both biases. To mitigate intensity bias, we integrate semantic perception and restoration modules that jointly perform image-level and feature-level denoising. We leverage the discrepancy between the restored features and the distorted features as a representation of distortion intensity. Subsequently, we employ intensity-aware cross-attention to adaptively weight the distorted features and discrepancy features for evaluating images with varying distortion levels. To address domain bias, we introduce a distortion domain recognition task and utilize the domain-wise similarity between synthetic and real distortions to weight the evaluation results from the two domains. Experiments on single datasets and cross-dataset tests demonstrate that our proposed method can better handle diverse distortion scenarios, achieving state-of-the-art performance.

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
