# OpenReview forum: "Towards Distortion-Debiased  Blind Image Quality  Assessment"
_acmmm.org/ACMMM/2024/Conference — MM2024 Poster_

### Official Review · Reviewer_nciU · 2024-05-09

**Rating:** 5
**Confidence:** 3

**Summary:**

The paper proposes a blind image quality assessment (BIQA) model that explicitly models distortion intensity and domain, reducing the impact of these biases on BIQA models. The proposed method mitigate intensity bias by using restored images as references and address domain bias through classification within the distortion domain. The experimental results indicate that the authors’ BIQA method achieves the best performance across various types of IQA datasets.

**Strengths:**

1. The authors have effectively demonstrated the impact of distortion intensity and domain on IQA model biases through a series of intuitive experiments and illustrations.
2. The experiments conducted in the paper are thorough and comprehensive, providing solid evidence of the proposed model’s efficacy.

**Limitations:**

The paper does not exhibit significant weaknesses; however, there are a few formatting issues that need to be addressed:

1. The Transformer model mentioned should be properly cited with the relevant references.

2. In Section 4.1, 4.2, 5.1 and 5.4, you should use *\paragraph* command. As per Section 9 of the official template: "Simulating a sectioning command by setting the first word or words of a paragraph in bold face or italicized text is **not allowed**".

3. Formula 3, 4 and 13 should be followed by a period to maintain consistency in punctuation.

4. In line 781, there is a "Table 3 3".

5. The format of the reference list should be consistent.

**Suitability:**

3

---

### Official Review · Reviewer_vE42 · 2024-05-22

**Rating:** 3
**Confidence:** 4

**Summary:**

In this paper, the authors propose a distortion-debiased BIQA method by integrating distortion perception and restoration modules. The restoration module uses a combined image-level and feature-level denoising method to restore distorted images, based on which the authors introduce a distortion domain recognition task, leveraging inherent differences between synthetic and authentic distortions for adaptive quality score weighting.

**Strengths:**

+ A new distortion-debiased BIQA method is proposed to handle synthetic and authentic IQA.
+ The experimental design is relatively complete.

**Limitations:**

- To address distortion intensity bias, the authors design an Image Restoration Module by performing image-level and feature-level denoising to represent distortion intensity. However, similar efforts have been around for a long time, such as Hallucinated-IQA [1] and AIGQA [2]. The author needs to explain the differences between them.

     [1] Hallucinated-IQA: No-Reference Image Quality Assessment via Adversarial Learning, CVPR.

     [2] Blind Image Quality Assessment With Active Inference, TIP.

- The distortion type and distortion intensity of authentically distorted images are relatively complex and usually contain multiple distortion types and unevenly distributed distortion intensity. In this case, how the Distortion Domain Classifier proposed by the author is implemented.

- Different image restoration models may obtain different results, and the author needs corresponding experiments to verify that the proposed framework is effective.

- As an IQA dataset in a real environment, experimental validation on SPAQ [3] also needs to be conducted. In addition, generalization experiments in other databases should also be added.

    [3] Perceptual Quality Assessment of Smartphone Photography, CVPR.

- The structure of the paper is not well organized, and the writing quality needs to be improved. The paper contains many errors of expression, and the language is stilted, making it difficult to follow.

- The format of References is not standard. For example, the name of the conference is sometimes abbreviated and sometimes not abbreviated, etc.

**Suitability:**

3

---

### Official Review · Reviewer_eBVa · 2024-05-22

**Rating:** 3
**Confidence:** 3

**Summary:**

This paper aims at address the challenge of distortion biases in IQA, and proposes a unified learning framework to effectively predict the quality of authentic images and synthetic images with different distortion intensities. This framework integrates distortion perception and restoration modules to address intensity bias. To tackle domain bias, this work introduces a distortion domain recognition task, leveraging inherent differences between synthetic and authentic distortions for adaptive quality score weighting.

**Strengths:**

The motivation for this work is sufficient, and the description of the method in this work is detailed.

**Limitations:**

1 In the introduction line 98, the authors mention that current research mostly focuses on constructing BIQA models tailored to specific distortion scenarios. However, as far as I know, DBCNN[1], UNIQUE[2], and DACNN [3] all aim at addressing the authentic and synthetic scenarios, although they do not design for the distortion intensities. I think the author should discuss the differences between your work and these mentioned works.
2 The related work is not focused enough, What is the purpose of discussing BIQA methods with recovered information, and BIQA methods with a single distorted image? I think the author should discuss the related work from the intensity bias perspective and domain bias perspective. This part lacks a discussion on the newest IQA work.
3 More visualization results need to be provided in Result to verify the effectiveness of distorted image reconstruction.
4 There are some typos in the manuscript, such as image eestoration module in line 360. Please check the manuscript more carefully.

[1]W. Zhang, K. Ma, J. Yan, D. Deng and Z. Wang, "Blind Image Quality Assessment Using a Deep Bilinear Convolutional Neural Network," in IEEE Transactions on Circuits and Systems for Video Technology, vol. 30, no. 1, pp. 36-47, Jan. 2020.
[2]W. Zhang, K. Ma, G. Zhai and X. Yang, "Uncertainty-Aware Blind Image Quality Assessment in the Laboratory and Wild," in IEEE Transactions on Image Processing, vol. 30, pp. 3474-3486, 2021.
[3]Z. Pan et al., "DACNN: Blind Image Quality Assessment via a Distortion-Aware Convolutional Neural Network," in IEEE Transactions on Circuits and Systems for Video Technology, vol. 32, no. 11, pp. 7518-7531, Nov. 2022.

**Suitability:**

2

---

### Official Review · Reviewer_2XcB · 2024-05-24

**Rating:** 3
**Confidence:** 3

**Summary:**

This work designs a distortion-debiased BIQA method to address both intensity bias and distortion domain bias. To mitigate intensity bias, the discrepancy between the restored and distorted features is used to represent distortion intensity, and a distortion intensity matrix is calculated for adaptive handling of intensity bias. In addition, a distortion domain recognition task is proposed to address domain bias, which utilizes the domain-wise similarity between synthetic and real distortions to weigh the evaluation results from the two domains.

**Strengths:**

The authors describe clearly the motivation of the proposed method and demonstrate its effectiveness. The ablation experiment is sufficient and reliable, verifying the effectiveness of each module.

**Limitations:**

1. There are many components and symbols in Figure 3, and the novel parts are not clearly highlighted.
2. The related work for the latest methods is not sufficient, besides, there are many universal IQA methods to assess both synthetical and authentical distortions. It is better to compare these methods in the introduction and related work to highlight the contributions.
3. In page 5, lines 550-552, “The IRM module is separately trained while other modules are optimized end-to-end on IQA datasets using mean squared error (MSE) loss, with frozen IRM parameters.” This sentence would be better written in the “Experimental Setups” section. It is necessary to check the organization of the content in the paper.
4. Most of the comparison methods in Table 1 are those in 2022. To prove the superiority of this method, it is suggested to add the latest method for comparison.

**Suitability:**

2

---

### Meta-Review · Area_Chair_nWbq · 2024-07-06

**Recommendation:** Accept (Poster)
**Confidence:** 3

**Metareview:**

The authors tackle distortion biases in IQA, proposing a unified learning framework that predicts the quality of authentic images and synthetic images with different distortion intensities. This framework integrates distortion perception and restoration modules to address intensity bias. The method is technically sound and is effective.

There are some issues with the paper that need to be fixed in the revision.  First, some details of the methodology need to be made more clear. Also, the contributions in relation to other references should be improved. Finally, the experimental validation is lacking in some way, in particular in terms of comparison with other methods.  The authors should make a strong effort in revising the paper, taking into considerations the various corrections and suggestions, to make strengthen the quality of the paper and its contributions to the research community.